Deep learning-based information retrieval with normalized dominant feature subset and weighted vector model

Eswaraiah Poluru
Syed Hussain hussain.syed@vitap.ac.in
School of Computer Science and Engineering, VIT-AP University , Amaravati , Andhra Pradesh , India
Kryvinska Natalia
Electronic publication date: 2024 Jan 22
Publication date: 2024
Volume: 10
Electronic Location ID: e1805
Received 2023 Jun 1; Accepted 2023 Dec 18
Copyright: ©2024 Eswaraiah and Syed
Copyright year: 2024
Copyright holder: Eswaraiah and Syed
License: This is an open access article distributed under the terms of the Creative Commons Attribution License, which permits unrestricted use, distribution, reproduction and adaptation in any medium and for any purpose provided that it is properly attributed. For attribution, the original author(s), title, publication source (PeerJ Computer Science) and either DOI or URL of the article must be cited.
License URL: https://creativecommons.org/licenses/by/4.0/

Keywords: Big data, Feature extraction, Feature subset, Feature selection, Feature vector

Funding: The authors received no funding for this work.

==============================
Multimedia data, which includes textual information, is employed in a variety of practical computer vision applications. More than a million new records are added to social media and news sites every day, and the text content they contain has gotten increasingly complex. Finding a meaningful text record in an archive might be challenging for computer vision researchers. Most image searches still employ the tried and true language-based techniques of query text and metadata. Substantial work has been done in the past two decades on content-based text retrieval and analysis that still has limitations. The importance of feature extraction in search engines is often overlooked. Web and product search engines, recommendation systems, and question-answering activities frequently leverage these features. Extracting high-quality machine learning features from large text volumes is a challenge for many open-source software packages. Creating an effective feature set manually is a time-consuming process, but with deep learning, new actual feature demos from training data are analyzed. As a novel feature extraction method, deep learning has made great strides in text mining. Automatically training a deep learning model with the most pertinent text attributes requires massive datasets with millions of variables. In this research, a Normalized Dominant Feature Subset with Weighted Vector Model (NDFS-WVM) is proposed that is used for feature extraction and selection for information retrieval from big data using natural language processing models. The suggested model outperforms the conventional models in terms of text retrieval. The proposed model achieves 98.6% accuracy in information retrieval.

Introduction

The goal of data retrieval is to discover the most relevant item in a data library based on a query. Cross-domain retrieval, particularly across picture and text domains, has recently gained in importance due to recent efforts. Generally speaking, there are two ways to approach the problem of image-text retrieval (Guo et al., 2015). There are a number of ways to evaluate visual-semantic embeddings, one of which is to match images to sentences or the other way around. Text-based retrieval, on the other hand, is described where an image is used as a query and a written command describes the intended adjustments to the image (Ahmad et al., 2018). Searches are performed by mapping queries and data items to a feature set in both cases. In spite of significant progress (Xia, Miao & Fan, 2020), basic frameworks for retrieval are based on an assumption that the embedding space information is well represented by either distance measurement or negative trigonometric distance, both assuming the features are in a higher dimension (Sun et al., 2020).

A geodesic proximity can better establish a link between objects under the statement that images lie on a low-dimensional manifold within a strong feature space (Fernandez-Beltran & Pl, 2018). There is also a problem with an under sampled feature space in the field of image processing due to the low amount of samples available (Yilmaz, Yazici & Kitsuregawa, 2014). An extremely sparse feature space makes it difficult to identify a correct neighbourhood for a given location and compute correct geodesic distances (Guo et al., 2018). Because these spots are regarded unrecoverable, it is found that the highest errors in typical retrieval techniques are considered for analysis (Liu et al., 2019a).

Text classification tasks with enormous dimensionality, including such sentiment analysis, emotions detection, and spam detection, necessitate the use of feature selection (FS) (Aygun & Benesova, 2018) in data mining. It is the goal of feature selection to pick from vast datasets only those features that are useful and informative (Biten et al., 2019). Because of this, FS can make machine learning algorithms more efficient while also reducing the size of the training space. Because of its impact on categorization accuracy, FS is seen as a key tool in this endeavour (Devlin et al., 2018).

Different pre-processing methods are the two main types of FS methods. Methods of the filter approach analyse the feature space statistically in order to choose a distinct subset of features (Faghri et al., 2018). To establish the quality of a feature subset, wrapper approaches use a research methodology to evaluate its performance by submitting it to a classifier and analysing the results (Fan et al., 2019). For each classifier, these two procedures are repeated until an appropriate quality feature subset is found. When the set of features is fairly tiny, wrapper methods (Goodfellow et al., 2014) are more efficient than filter methods, but they have a high computational complexity (Hardoon, Szedmak & Shawe-Taylor, 2004). Contrary to this, filtering approaches are fast, flexible, and do not require any interaction with a classifier as they are being used to build the feature set (Elfaik & Nfaoui, 2020). Due to classifier interaction, the FS approach may only be useful for a specific learning algorithm, increasing the execution time. Consequently, huge datasets benefit from filter approaches (Huang & Peng, 2018).

Global features explain the overall visual content of a feature set, local features describe the visually relevant information present in a small, nearby part of the feature set. Fusions of local and global characteristics are often chosen for information retrieval as they boost performance. Features that are taken from the entire feature set are called global features. On the other hand, local features are gleaned from similar sub features. As a whole, a global feature can be generalized by a single vector. This makes them easy to implement into existing classification schemes. The computation of local features, on the other hand, occurs at numerous feature locations, making them less susceptible to artifacts. In addition, as they are centered on the sequential programming model, most existing FS approaches for text categorization (Huang, Long & Wang, 2019) are filter-based yet ineffective when datasets are huge. Data must be read into storage for analysis by standard FS algorithms, but a limited storage cannot handle big datasets. The filter FS technique for text classification is depicted in Fig. 1, feature selection process as a distributed process.

Figure 1 Feature selection process.

A lot has been accomplished in the area of cross-modal retrieval in the previous ten years. Multimodal retrieval seeks to find relevant information from a variety of sources. In several disciplines, it is frequently employed, such as visual set of questions, picture or video description, phrase localisation (Huang & Wang, 2019), transfer of knowledge and message synthesis. In a single mode, the system could just about bridge that semantic gap between elevated human understanding and low-level features with natural language processing (NLP) approaches (Ji et al., 2019). Researchers are looking into the conceptual gap between images and text as deep learning produces outstanding achievements in both the vision and language domains. Image-to-text retrieval and text-to-image retrieval are terms used interchangeably in unidirectional image-text cross-modal retrieval (Jian et al., 2019). Images, videos, and audio are all included in the overviews of audiovisual information retrieval. Data retrieval for music and sound has been the focus of a recent cross-modal overview. Even though NLP has lot of ambiguity, the proposed model follows a strategy that clearly and efficiently performs the information retrieval. Initially the records in the dataset are analyzed and then the feature extraction is performed. The extracted features undergo feature selection model that selects the relevant features and then normalization is applied on the selected features and the relevant information is extracted based on the features.

Unstructured or semi-structured data can be benefited from the process of information extraction (IE) (Lee et al., 2018). With the fast expansion of multifarious, sometimes known as multidimensional, unstructured data, big data presents new issues for IE approaches. Unstructured massive data overwhelms traditional IE systems, making them unable to handle it effectively (Li et al., 2017). The sheer volume and variety of big data necessitates an upgrade in the IE systems’ computational power. IE approaches for data pre-processing, data retrieval and conversion, and representations for large amounts of multidimensional unstructured data need to be evaluated for their capabilities and limits. The general process of information extraction by feature subset is shown in Fig. 2, feature extraction procedure in vectorization.

Figure 2 Feature extraction.

The query is what a search engine sees as the user’s entry into the system. Boolean expressions over content keywords are the most common type of query, but complete or partial documents can also be included. Boolean expressions are commonly used as a way to retrieve information from natural language queries in some systems (Li et al., 2019). Boolean inquiries, on the other hand, are not appropriate for describing the information demands of consumers. The relative relevance of each term cannot be indicated, which makes them difficult to arrange for inexperienced users (Lin et al., 2014). Many programs allow the query procedure to be iterative in order to solve these issues. As a starting point, a user’s initial inquiry is used as a starting point for further refinement of the user’s information demand (Mao et al., 2019).

There are further advantages to using a distributed storage and processing model for large amounts of multimedia data (Liu et al., 2019b). The storage and processing of multimedia large data is a problem even though multimedia computing alleviates some of the computational load and maintenance issues (Chen & Bazzani, 2020). As more and more multimedia documents are created, stored, processed, altered, and retrieved, the semantic information in the underlying storage systems and data formats becomes more and more identical. Heterogeneity is the most common restriction that affects multimedia large data storage and retrieval. User intent is another complication in multimedia retrieval. The retrieval process is usually constrained to a consistent framework of multimedia data like photos in most existing approaches. Lack of type variety reduces the quality of service (QoS) of multimedia retrieving since it is difficult to detect the intent of users’ searches. The key contributions of the proposed method are as follows.

• In this study, we introduce a novel approach termed the Normalized Dominant Feature Subset with Weighted Vector Model, or NDFS-WVM.

• Our method begins with feature extraction using a weighted vector technique. Following feature extraction, we perform two crucial steps: feature normalization and feature weight assignment.

• Ultimately, our approach culminates in the generation of a normalized dominant feature vector after undergoing feature processing within hidden layers.

This section briefly discusses about the information retrieval process and feature extraction model for exact information retrieval based on query. The section ‘Literature Survey’ discusses literature survey on numerous models for information retrieval. The ‘Proposed Model’ discuss about the proposed model and the algorithm for information retrieval using dominant feature subset. The ‘Results’ provides the results section and ‘Conclusion’ concludes the article.

Literature Survey

The irregular structure and high degree of detail in the documents make computerized information retrieval and the idea of text summary a challenging procedure in natural language processing. The technique of summarizing produces a summary by rephrasing a lengthy text. Earlier approaches for information retrieval and distillation focused on the specific sub-domain, used a large labeled dataset, and relied on domain-specific knowledge to increase performance. In this research, Mahalakshmi & Fatima (2022) provided a novel text summarizing technique built on deep learning (DL). Data retrieval, template generation, and text summarization are the three main components of the proposed paradigm. To begin retrieving the textual material, the bidirectional long short term memory (BiLSTM) method is used; this method makes an assumption about each word in a phrase, extracts relevant information from it, and embeds it into a semantic vector. The next step is to generate templates with the DL model. For the purpose of summarizing the text, a deep belief network (DBN) model is used. For the entities that can be seen in the photos, a description of those images is also provided. The work’s originality is demonstrated by the integration of BiLSTM and the DBN model during the text summarizing and image captioning processes.

Semantic-based Heterogeneous Multimedia Retrieval (SHMR) was created by Guo et al. (2015) employed cheap cost for retrieving and storing semantic information. Uniqueness in big data environments including heterogeneous multimedia retrieval was initially addressed by this concept, which has since evolved. New methods for obtaining and representing semantic information were then proposed. The NoSQL-based architecture used in this model also provided semantic storage for multimedia data, which was then processed in parallel across distributed nodes. A user feedback-driven method and MapReduce-based retrieval technique were used to produce good quality of service and high retrieval precision in the proposed model. In terms of both economic productivity and retrieval performance, the proposed model is superior.

Bi-Directional Fast Fourier Transform (BD-FFT) was developed by Ahmad et al. (2018) to extract the compressed binary codes from the high-dimensional deep features. It served as a memory saver and a computation simplification for fast data retrieval. Approximate nearest neighbor (ANN) search methods were used to find images in large datasets using the altered codes. Based on experiments with convolutional neural network (CNN) characteristics, it has improved retrieval accuracy by experimenting with lower length codes. The proposed retrieval model was tested on seven large-scale datasets in order to verify its effectiveness.

The ability to quickly and accurately retrieve relevant medical images is crucial to the practice of modern medicine. In this research, Iqbal et al. (2023) provided a novel approach to medical picture methodology selection by integrating textural and visual data. Because it can help with image examination and provide vital details regarding the imaging technique employed, understanding the imaging process behind a concept like a chest X-ray, skin dermatology, or mammary histopathology image can be very useful for healthcare practitioners. To do this, the author utilized the visual and textural features of medical images through deep learning-based feature engineering. Using a predetermined CNN, the author was able to extract high-quality visual data from the photos. Physically separating the relevant elements from the other visual and factual features of the images is achieved by the use of the Global-Local Pyramid Pattern (GLPP), Zernike moments, and Haralick. Details regarding the technology can be gleaned from crucial qualities including picture modality and imaging technique specific properties. To merge the textural and visual components, the author used a feature fusion technique that takes into account the depictions acquired from the two modalities. By increasing the discriminative power of the feature vectors by fusion, accurate modality classification is made achievable.

Wang & Ulukus (2022) looked at the SPIR issue from the perspective of user-side common randomness. In Symmetric Privacy-Preserving Information Retrieval (SPIR), a user retrieves a message from a set of K messages stored in N non-colluding and replicated records in such a manner that every database understands the retrieved message index and the user learns nothing more than the retrieved message. The three characteristics of SPIR are that it is impractical for a single database, that it requires shared common randomness among the databases, and that its capacity is lower than that of PIR, which requires merely user privacy. The author presented a new SPIR variation in which the user is given access to some portion of the common randomness of the databases that they share. The author pinpointed the precise capacity region of the triple (d, S, U) where d is the download cost, S is the amount of common randomness available on the shared database, and U is the amount of common randomness available on the user’s side.

Heidarzadeh & Sprintson (2021) investigated the function of encrypted off-server data in PIR on a single server. A user wants to get a message from a server that maintains K messages that are dispersed uniformly and independently. This is an example of the single-server PIR problem. The author thought about scenarios where the user is given a linear assortment of some subset of M messages from the database as part of the coded side information. In this scenario, both the server and client start off in the dark about the real identities of the M messages that make up the support set for the coded side information. The author looked at two models, one where the requested message is guaranteed to be in the support set of the coded side information, and another where it is not. The author also took into account the following two privacy concerns: Either (i) it is necessary to conceal both the demand and support sets of the coded side information, or (ii) it is sufficient to conceal only the demand set. The author explored the problem of building a protocol for producing the user query and the server’s answer that allows the user to decode the necessary message while satisfying the privacy requirement for each model and each privacy criterion. All (scalar-linear) protocols that meet the privacy criterion are characterized by our definition of their (scalar-linear) capacity, which is the ratio of the number of binary bits in a message to the minimum amount of information bits retrieved from the server.

A Levinberg-Marquard algorithm-based Deep Typical Correlation Analysis approach (LM-DCCA) based on deep correlation mining was created by Xia, Miao & Fan (2020). This model’s goal was to use the provided algorithm to train a variety of medial characteristics. Finally, the correlation between learned characteristics was found so that different multimedia data could be compared. Local optima problems have been handled using the Levenberg–Marquart approach with this strategy. Cross-media retrieval was tested on a variety of datasets to demonstrate its effectiveness. When compared to other advanced multimedia retrieval models, this model has proven to be superior.

Sun et al. (2020) implemented a new graph categorization semantic similarity model by reducing features. In this case, neural language systems with subtree patterns were used to learn the representations of vectors. The new features were created by merging semantically comparable subtree patterns. For the purpose of picking the most discriminative features, this model has provided a new feature discrimination score. The proposed retrieval approach, which makes use of the proposed FRS-KELM graph classifier, was put to the test using real datasets.

Big data challenges and data reduction were the focus of Fernandez-Beltran & Pl (2018) when they created a novel multimedia retrieval model based on probabilistic latent semantic analysis (pLSA). The goal of this approach was to improve the extraction process for content-based multimedia retrieval (CBMR). It has been decided to employ pLSA and latent dirichlet allocation (LDA) in CBMR to discuss the results of document reduction rather than other existing methodologies. The retrieval performance of CBMR has been incorporated in the proposed model.

Yilmaz, Yazici & Kitsuregawa (2014) created a RELIEF-based modality weighting technique named RELIEF-MM technique. Unbalanced datasets, multi-labeled data and noise, and class-specific feature selection were some of the issues that had to be overcome in order to implement this method. Reliability and representation capacities of modalities can be demonstrated using RELIEF-increased MM’s weight estimation function, which is less computationally intensive. TRECVID 2007, TRECVID 2008, and CCV datasets were used in the experimental investigation to demonstrate the resilience and accuracy of the modality weighting model for multimedia data retrieval.

To store and retrieve ontologies using big data techniques, Guo et al. (2018) proposed a new heterogeneity multimedia data retrieval approach based on Semantic Ontology Retrieval (SOR). Ontology representation and semantic extraction on multimedia big data were initially solved. For model definition, SOR was employed as a retrieval mechanism. It was also utilised to develop a new retrieval system decentralized distributed nodes using MapReduce for concurrent processing of SOR. One way for improving the overall user experience and increasing the accuracy of search results was also discussed. It was found that the SOR intended for heterogeneous multimedia and semantic-based retrieval was a good fit for the task at hand.

By combining deep hashing and multiple levels of semantic similarity learning, Liu et al. (2019a) developed a method for discovering multilevel similarity measure correlations of multimedia data. Unified binary hash codes were first learned to investigate multi-level correlations by exploiting semantic labelled data and local structure. As a result, limitations like quantization and bit balancing were used to reduce the size of the unified hash codes. This model uses a combination of deep neural networks (DNNs) and learnt unified binary codes to learn feature representations. With the new model, prediction errors in network and unified binary code outputs were reduced. Both text- and image-query-and-image-task simulation results were employed on two widely used multimodal collections to demonstrate the effectiveness of the approach.

An image retrieval approach that relies on color-shape feature extraction methods was presented by Faghri et al. (2018). Colour histograms and discrete wavelet transforms (DWT) are used in HSV colour space to extract colour characteristics, however EHD is utilised for texture since EDH is more efficient at discovering relevant images when it is used in MPEG-7, which uses DWT. The Corel dataset, the most frequent dataset in the CBIR sector, was used to evaluate this retrieval system. A machine learning technique is considered a severe limitation, but the method has superior efficiency in terms of accuracy than other state-of-the-art approaches.

Sentiment analysis can accurately forecast if an item will have a favorable, negative, or neutral tone. Data mining, computational linguistics and natural language processing are among the fields that it impacts. Emotions can be expressed openly or covertly. Analyzing sentiment in Arabic poses challenges because to the language’s diverse morphology, limited resources, absence of context, and the lack of unambiguous sentiment expressions in implicit text. Deep learning has emerged as the most effective approach for conducting sentiment analysis on Arabic text. Nevertheless, there is a requirement for improvements in the existing level of precision in assessing Arabic emotion, taking into account both the statement’s context and the underlying attitude conveyed. The study conducted by Hanane Elfaik & Nfaoui (2020) explores the application of a bidirectional long short-term memory (BiLSTM) Network in improving Arabic Sentiment Analysis. The BiLSTM network is utilized to capture and retain memories in both forward and backward directions.

In this article, Li et al. (2019) introduced Object-driven Attentive Generative Adversarial Networks (Obj-GANs) to aid in the synthesis of images based on text descriptions that focus on specific objects in complex circumstances. A novel object-driven attentive image generator is proposed, which utilizes a pre-generated semantic layout and extracts the most relevant phrases from the textual description to synthesize prominent objects. Furthermore, a novel Fast R-CNN object-wise discriminator is proposed to provide comprehensive object-wise discriminating signals by assessing the alignment between the synthesized object and the text specification, as well as the pre-generated layout. The suggested Obj-GAN demonstrates significant advancements compared to the previous state-of-the-art on the extensive COCO benchmark. Specifically, the Inception score shows a 27% improvement, while the FID score decreases by 11%. To gain a deeper comprehension of how the suggested model produces intricate visuals of superior quality, we conduct a comparative analysis between the traditional grid attention and the novel object-driven attention. This involves deconstructing their mechanisms and illustrating their attention layers.

The subjective technique of CBIR was developed by Hardoon, Szedmak & Shawe-Taylor (2004), utilizing a mixture of low-level information. Color characteristics were derived from the HSV color model, while texture characteristics were derived using the DWT and Gabor wavelet. A supplementary attribute measuring 1,250 units was calculated and included in the feature set for color and edge directionality. Increasing the dimension of the feature vector leads to lengthier search and comparison times, but it also improves the accuracy of the findings. The recommended approach demonstrated excellent average precision and recall when assessed across diverse datasets. The suggested system lacks texture and spatial information.

Images can be searched based on colour, form, and texture, according to a new technique developed by Huang, Long & Wang (2019). Color data were extracted using the Canny edge distribution and the DWT morph in the YCbCr colour space, while texture characteristics were extracted using the GLCM transform. The RGB color space had been used to recover shape characteristics using the canny edge approach. There was an attempt to improve the quality of the answer by using genetic algorithm (GA) in conjunction with simulated annealing (SA) model. The suggested CBIR system has better average precision and recall than other current systems. Because the cooling process requires so many repetitions, SA’s computation time is significantly longer than that of other methods. The proposed study introduces a new approach for the Normalized Dominant Feature Subset with Weighted Vector Model, based upon existing literature evaluations (refer to Table 1).

Table 1 Summary of literature reviews.

Author names	Year of publication	Manuscript title	Proposed model	Limitations	
Guo et al. (2015)	2015	An effective and economical architecture for semantic-based heterogeneous multimedia big data retrieval	The author provided a novel text summarizing technique built on deep learning (DL). Data retrieval, template generation, and text summarization are the three main components of the proposed paradigm.	The heterogeneous big data contains lots of noise that impacts the accuracy levels. The preprocessing model needs to be applied for better precision rate.	
Xia, Miao & Fan (2020)	2020	A cross-modal multimedia retrieval method using depth correlation mining in big data environment	A Levinberg-Marquard algorithm-based Deep Typical Correlation Analysis approach (LM-DCCA) based on Deep Correlation Mining was created by the author.	The model retrieval rate is high however, the time a complexity level of the model is high that need to be reduced.	
Sun et al. (2020)	2020	Feature reduction based on semantic similarity for graph classification	The author implemented a new graph categorization semantic similarity model by reducing features. In this case, neural language systems with subtree patterns were used to learn the representations of vectors.	The feature reduction model calculation computational complexity levels are high. The proposed model similarity check is time consuming and minute change in attributes will be discarded that needs to be avoided.	
Fernandez-Beltran & Pl (2018)	2018	Prior-based probabilistic latent semantic analysis for multimedia retrieval	Big data challenges and data reduction was the focus of the author when they created a novel multimedia retrieval model based on probabilistic Latent Semantic Analysis (pLSA). The goal of this approach was to improve the extraction process for content-based multimedia retrieval (CBMR).	The error rate of the model is high that need to be reduced. The probabilistic model can be enhanced with the gradient models for better efficiency.	
Yilmaz, Yazici & Kitsuregawa (2014)	2014	RELIEF-MM: effective modality weighting for multimedia information retrieval	The author created a RELIEF-based modality weighting technique named RELIEF-MM technique. Unbalanced datasets, multi-labeled data and noise, and class-specific feature selection were some of the issues that had to be overcome in order to implement this method.	The problem regarding existence and existential assumptions of big data is high in this ontology model that can use maximum likelihood probability for better accuracy rate.	
Mahalakshmi & Fatima (2022)	2022	Summarization of Text and Image Captioning in Information Retrieval Using Deep Learning Techniques	The author provided a novel text summarizing technique built on deep learning (DL). Data retrieval, template generation, and text summarization are the three main components of the proposed paradigm.	The hidden layer for processing for the models is high that is used for the kernel layer filtering. The kernel filtering size can be increased for accurate filtering and to reduce the error rate.	
Iqbal et al. (2023)	2023	Fusion of Textural and Visual Information for Medical Image Modality Retrieval Using Deep Learning-Based Feature Engineering	The author provided a novel approach to medical picture methodology selection by integrating textural and visual data. Because it can help with image examination and provide vital details regarding the imaging technique employed, understanding the imaging process behind a concept like a chest X-ray, skin dermatology, or mammary histopathology image can be very useful for healthcare practitioners.	The proposed model fusion process time complexity is high and the error rate is also observed high. The feature processing is identified as more complex that can be reduced with dimensionality reduction.	
Wang & Ulukus (2022)	2022	Symmetric Private Information Retrieval at the Private Information Retrieval Rate	The author looked at the SPIR issue from the perspective of user-side common randomness. In Symmetric Privacy-Preserving Information Retrieval (SPIR), a user retrieves a message from a set of K messages stored in N non-colluding and replicated records in such a manner that every database understands the retrieved message index and the user learns nothing more than the retrieved message.	The iterations performed in information retrieval process experiences computational complexity that can be reduced with the hybrid optimization models.	
Heidarzadeh & Sprintson (2021)	2021	The Role of Coded Side Information in Single-Server Private Information Retrieval	The author investigated the function of encrypted off-server data in PIR on a single server. A user wants to get a message from a server that maintains K messages that are dispersed uniformly and independently.	The delay levels in the model for information retrieval is high that can be reduced using the deep learning models that has high end hidden layer processing models.	

Research gap

• In order to evaluate the model’s performance when faced with noisy or unstructured text data, such as user-generated content, which often contains misspellings, slang, or incomplete sentences.

• It is difficult to investigate the model’s scalability and efficiency in handling extremely large text datasets, such as those encountered in real-world applications like web-scale search engines or social media platforms.

• The examined models exhibits extended processing times and tend to retrieve irrelevant text, which requires mitigation.

• Several models exhibit high computational complexity, leading to increased model latency. There is potential to decrease the number of iterations and expand the amount of processed text to enhance overall performance.

Proposed Model

Retrieving documents that are relevant to a user’s search query from a big collection is the goal of information retrieval. Text is now treated as little more than a collection of words by statistical approaches, which are currently the most effective general purpose retrieval methods. More complex linguistic processing has been tried, but so far it has not had much of an impact on retrieval performance. Such processing can, in fact, reduce the retrieval’s effectiveness if it is not done with care. Compound index phrases are notoriously difficult to weight properly, and this makes it tough to improve upon a strong statistical baseline. There are also issues with implicit linguistic processing inherent in statistical methodologies.

When the weighted input exceeds a threshold, the output out from text document feature subset is converted to a binary value using activation functions, which returns 1. Otherwise, the value is 0. Default weights and biases are set to zero at the start. The correctness of a neural network’s output is all on finding the best values for biases and weights by constantly adjusting them. In recent years, the demand for automated text or document retrieval has grown significantly, and this has drawn the attention of NLP researchers (NLP). Document retrieval differs from data retrieval and question answering in that it aims to identify the most important characteristics of the information being sought, as well as to review previous experience in the field, as well as external developments that are igniting growing interest in document and text retrieval. Figure 3, show that hidden layer representation depicts the hidden layer architecture for feature processing.

Figure 3 Hidden layer representation.

A neural network’s hidden layers enable it to decompose its complex task into more manageable data-processing phases. The function of each hidden layer is optimized for a certain result. Between the input layer and the output layer is a set of hidden layers that does the actual non-linear processing of the data. These layers are two dense layers, two batch normalization layer and ReLU activation function used multiple times because the batch normalization used to normalize the layers are to get the feature faster and accurately the dense layer receives the neuron information from the previous layer to get appropriate features based on the query with correlation. To generate an output, a neural network technique routes its inputs through an activation function after giving weights to the inputs in a hidden layer. The network’s inputs are transformed in a nonlinear fashion by the hidden layers. The functions of a neural network’s hidden layers and weights vary greatly from one job to the next. Mathematically, obfuscation techniques fall within the category of squash functions. Because they take an input and return a value that falls inside the range for defining probability, these functions comes in handy when the expected result of the operation is a probability. The optimal range for the number of buried neurons is somewhere between the size of the input layer and the size of the output layer. The amount of features necessary to compute each hidden state, however, is independent of the length of text sequence.

The system receives documents as input. The majority of information retrieval systems construct an inverted file, or set of words in alphabetical order, even though tagging and information extraction can take place at this stage as well. This list excludes stopwords. To ensure that the system contains all of a words occurrences in one location, new documents are interfiled into the current list. More and more text retrieval systems include or develop knowledge bases with in-built internal lexicons and semantic networks or lists of expressions, synonyms, and pronouns of a certain person. Currently, multiple systems extract supplementary information or perform various operations on the words during storage. Here are a few examples: The analysis of the user’s text involves determining the stem of the word, identifying its part of speech, determining if it is a proper noun, and maybe analyzing its relationship with other words in a section or text as a whole. Some users may also automatically assign indexing to phrases or broad subject areas. A few NLP systems generate and preserve a representation of every sentence, encompassing the role and relationship of each word and phrase inside it. Currently, statistical systems calculate the weights.

As soon as users submit a request, the system’s response must be interpreted. Because the searcher has already stated the question in computer-interpretable words, this is a simpler task in Boolean systems than it would be in full NLP systems. NLP and statistically-based systems have to accomplish some of the work that searches do in preparing a question. The terms to be searched for are determined by a statistical algorithm, which may check for stems as well as singular and plural variants of words. Each phrase could be given a different weight. Objects, subjects, agents, and verbs can all be tagged by a comprehensive NLP system, which can also provide synonyms and other forms for proper nouns and other parts of speech. The system can then compare the query’s unambiguous representation with its knowledge base. Stems and fundamental syntax may be identified by a partial NLP system before query representation is created. The process of feature extraction and selection using weighted vector is shown in Fig. 4, weighted vector representation.

Figure 4 Weighted vector representation.

In this study, feature selection is a vital pre-processing step since it takes into account the most crucial aspects of the feature collection. Feature selection is the process of narrowing down a large set of candidate features to just those that will be useful, using an evaluation metric known as the correlation factor. Let FS be the larger feature set, consisting of features {F1, F2... FM,} where M is the total number of features. When given a set of T ≥ D characteristics to choose from, the goal of feature selection (FS) is to zero in on the G, most discriminatory ones.

The feature selection process and removal of irrelevant features are performed as (1) CorrSetM=∑Xi−X ¯Yi−Y ¯∑Xi−X ¯2 ∑Yi−Y ¯2

Here X i is the x number of samples of feature attributes, Y i is the y number of samples of feature attributes. X ¯ isthe mean of attributes in x number of attributes and Y ¯ is the mean of attributes in y number of attributes.

It is possible to incorporate natural language processing at any or all of these phases, depending on user preferences. For both the query and the document, full NLP understands and saves meaning at all phases and all levels. Practical concerns, such as how computationally intensive the additional processing will be, guide the decision of how much and where to add in any computer system. At this point, NLP has two clear advantages. As a first step, using real language allows us to convey actual information demands, as well as intentions and meaning, more effectively. False drops and other wrong word/wrong meaning retrievals can be eliminated when full NLP is used. Using NLP, queries can be narrowed down without omitting potentially useful information. Recall and accuracy are expected to improve. Synonyms, alternative forms, and geographic terms connected to a query can all be added using NLP. The most relevant text features can be automatically trained in a deep learning model using massive datasets including millions of variables. In this research feature extraction and selection model for information retrieval is proposed to perform retrieval from huge data utilising natural language processing models. The proposed model performs normalization of features for performing feature scaling. The purpose of normalization is to make all of the features uniform in size. The model’s performance and stability throughout training are both enhanced as a result. Feature Scaling was implemented with mean normalization. Mean normalization is a technique that takes all features and subtracts their mean value.

Experts manually identify the optimal region of a dataset for analysis, and this region is called the region of interest (ROI). A region of interest (ROI) is a defined area inside a dataset that will be the focus of an intended transformation. A region of interest can be represented by a binary mask. Pixels in the mask picture that are part of the ROI are assigned a value of 1 and those that are not are assigned a value of 0. Most algorithms take one of two main techniques, feature-based or object-based. Pixels that have substantial optical similarities to the target are grouped together using feature-based approaches to create ROIs. On the basis of optical feature similarity, these techniques can effectively capture most of the target pixels. However, not every pixel in the target has robust optical properties, thus the ROI that is recognized rarely covers the whole thing. The proposed model framework is shown in Fig. 5.

Figure 5 Proposed model framework.

The suggested model begins by taking the text dataset as input for processing. Feature extraction is conducted on the dataset, taking into account all the retrieved features. Feature normalization is a technique that transforms numbers from a high range to a low range, making them easier to process. Weight allocation is conducted for each feature, taking into account the correlation factor. The resulting weighted features are subsequently fed into the hidden layer for processing. The weight allocation is determined by evaluating the vectors with a low normalized range. A window size of 6 × 6 is employed for kernel processing, whereas a kernel size of 16 × 16 is utilized in the fully linked layer to generate the final feature vector. The ultimate feature vector is employed for the processing of information retrieval. When users submit a request, it is crucial to interpret the response returned by the system. Given that the user has already formulated the query in a format that can be understood by computers, employing Boolean systems is more efficient for completing this work compared to using full Natural Language Processing (NLP) systems.

Dataset input: The dataset is considered from a public service provider. The dataset contains records that are gathered from numerous social media applications and the posts of the users are considered.

Feature extraction: The goal of feature extraction is to reduce the size of a dataset by extracting relevant features from it and rejecting the rest. These updated reduced set of attributes should then be capable convey most of the information included in the original assortment of features. The original set of features can be combined in this fashion to provide a streamlined version of the original set. The process of extracting useful features from a dataset is known as feature extraction, and it is essential to the further stages of machine learning, such as training and generalization. The proposed model extracts all the features and these features are used for processing to create a feature vector.

Feature normalization: In machine learning, normalization is a scaling technique used in the data preparation phase to convert the numerical column values to a standard scale. Not all models need to include every possible dataset. Only when features of machine learning models fall into varying categories does this step become necessary. Normalization is a transformation technique that helps to increase the performance as well as the accuracy of your model better. When you don’t have a good idea of how your features are distributed, normalizing your machine learning model can help. Data features do not follow a Gaussian (bell curve) distribution, to put it another way. Outliers in data will be impacted by normalization because it requires a continuous range. The proposed model considers feature normalization that is used for feature engineering.

Feature weight allocation: In machine learning, normalization is a scaling technique used in the data preparation phase to convert the numerical column values to a standard scale. Not all models need to include every possible dataset. Only when features of machine learning models fall into varying categories does this step become necessary. Normalization is a transformation technique that helps to increase the performance as well as the accuracy of the model better. Data features do not follow a Gaussian distribution, to put it another way. Outliers in data will be impacted by normalization because it requires a continuous range.

Hidden layer processing: The raw data from the input layer is processed in the hidden layer. After this, the result is passed on to the output layer, where it will undergo further processing before being used to generate the final output. Between both the input and the output of a neural network method is a hidden layer, in which the function assigns weights to the inputs and routes them via an activation function to produce the desired output. In short, the hidden layers conduct nonlinear changes of the inputs inputted into the network.

Dominant feature vector generation: The proposed model generates the feature vector that is used for text processing and exact information retrieval. The feature normalization reduces the attribute range and the correlation model is used to select the minimum correlated set that is used for the detection of features in the feature vector. The dominant feature vector set is used for training the model.

Both numeric and categorized information can be found in the datasets. The string data formats used to describe categorical properties facilitate human comprehension. Categorical data is difficult for machines to interpret. For this reason, it is necessary to transform the category information into numeric form before proceeding with the analysis. The text can be transformed into a numerical format via count vectorization. The name “Document Term Matrix” is given to this outcome. Unique, significant words from the entire text are displayed in the columns. Each row shows how often that term appears in each phrase.

The inability of feature-based approaches to differentiate between targets introduces additional complexity into the processing pipeline. Any proposed feature extraction approach must capture the prominent aspects of the ROI without knowing already what those salient features would be in order to achieve the requisite classification accuracy. That is to say, any proposed feature extraction method, in whatever shape it takes, should preserve as much useful data as feasible. The proposed model considers feature based ROI to extract the data for accurate information retrieval. The main features are considered and the query based features are extracted from the dataset.

Algorithm NDFS-WVM

{

Input: Information Document Dataset {IDDSet}

Output: Normalized Dominant Feature Subset {NDFSset}

Step-1: Initially load the dataset and analyse the records from the IDDset. The record loading from the dataset is performed so that the information retrieval is performed from the big data. Each record contains information about the topic and the attributes are considered for processing. The record analysis is performed as (2) RecIDDseti= ∑i=1Mgetvaluerecordi+maxValuerecordi−μ

Here µis the null and special symbols wrongly indicated in the dataset that need to be removed, M is the total records in the dataset.

Step-2: The feature extraction process is performed from the dataset and the features are represented in a feature vector that is used for the accurate information extraction training. Each feature has its individual properties in the text document. The extracted features represent the text properties. The feature extraction process is performed as (3) Feij= ∑i=1M maxcorrii− minattri+λ×logKL′

(4) Fseti= ∑i=1M maxFei−mincorrFei+ ∑i=1MGL

Here corr() is used for identification of values related to neighbour record values. K′ is the features that are highly correlated. L is the features that are less correlated. G is the model used to find the standard deviation that is used for information extraction.

Step-3: The proposed model can handle dynamic range of normalized feature vector. The proposed model calculates the correlation factor initially and then the normalization is applied on the reduced feature set that has no static size. The feature normalization is performed to balance the attribute values in the dataset. The term normalization refers to the method used to structure a database. The process involves constructing tables and relating them to one another in accordance with principles meant to secure the data and increase the database’s adaptability by doing away with duplication and inconsistencies in dependency. The feature normalization helps in achieving high correlated values. The feature normalization is performed as (5) Normi= ∑i=1MLogminFsetiPi×Pi+1+max∑i=1Fseti ∗corrFseti,Fseti+1−L

Step-4: After normalizing the features, the weights are allocated to the features that are highly correlated. The weights are allocated to the features that are mainly used to train the model. Feature selection entails picking and choosing which of features to prioritize. The features you’ve chosen are ranked in order of importance to the model, and these rankings serve as the weights for the features. The term feature weight is shorthand for category weight, which is the calculated value assigned to each feature item in the feature set based on the weighting procedure. The weights are allocated to the features after normalization. The weight allocation is performed as (6) Fei ∑Normi+maxFseti+K∑i=1Normi−L

Step-5: The initialization of the hidden layer model is performed that considers the weighted features. The input is represented as L and the hidden layer is indicated as Hi, sigmoid activation function is represented as σ, weights W, Bias B, Input and output weights Wi and Wo, input and output bias is indicated as Bi and Bo and output is represented as Ot and error is indicated as Err. A typical CNN will have several hidden layers, including convolutional layers, pooling layers, fully connected layers, and normalizing layers. The term convolutional and pooling functions are being used in place of the “normal” activation functions defined above. Based on the parameters, the feature processing is performed as (7) Pros=σLiWi+LrBi+Kr←Wo

(8) WPros=σLiWi+LrBi+Kr←Ot

(9) Fpros=σWiBi+Bo+ maxWPros+Ki+1Oi+ minErr−minPros

Step-6: The proposed model is designed mainly for information retrieval on large volumes of data. The query text is processed and then relevant features are extracted, normalized and weights are allocated and based on that the information is extracted. The dominant feature subset model is generated from the extracted normalized feature subset. The dominant feature subset is generated that is used for accurate information extraction from the document. The normalized dominant feature vector set is generated as (10) DFVseti=KiM+FprosWi+maxcorrFprosWo,Ot∑i=1N maxWi+minBi

}

Results

The selection of text feature items is crucial for text mining and information retrieval. Conventional methods of feature extraction require carefully prepared features. Learning new effective feature representations from training data for new applications is made possible by deep learning, but it would take a long time to hand-design such a feature. The field of text mining has made great strides because to the advent of deep learning, a novel feature extraction tool. Traditional methods, on the other hand, rely on designers’ prior understanding, meaning they can’t make use of large amounts of data. In contrast, big datasets are automatically scoured by deep learning to learn features automatically. Deep learning allows for the automated learning of feature representations from enormous datasets that can contain billions of parameters.

The suggested model is coded in Python and run on Google Colab. The dataset can be accessed at the following URL: https://www.kaggle.com/datasets/dmaso01dsta/cisi-a-dataset-for-information-retrieval. The proposed model uses Centre for Inventions and Scientific Information (CISI), a big data dataset for Information Retrieval. The information consists of 1,460 documents and 112 queries and was compiled by the CISI. It is designed to be used in information retrieval models to generate a list of document IDs that are pertinent to a particular query. To evaluate the model’s efficacy, users can compare it to the gold standard list of query-document matching included in the file “CISI.REL.” This list serves as the ground proof against which our model can be evaluated.

An information retrieval system utilizes a database of natural language documents to locate the precise documents that provide the response to a user’s query. Library systems are the origin of these innovations. Nevertheless, these technologies just facilitate users in finding the necessary data, without generating or offering any answers. It notifies the user about the existence and accessibility of documents that might have the desired information. Relevant papers are defined as those that precisely fulfill the requirements of the user. An optimal information retrieval system will exclusively provide pertinent materials. The present study introduces a method called Normalized Dominant Feature Subset with Weighted Vector Model (NDFS-WVM), which is utilized for extracting and selecting features in order to retrieve information from large datasets utilizing natural language processing models. The new model is evaluated against the existing BiLSTM-DBN model (Elfaik & Nfaoui, 2020) and the RELIEF-MM model (Yilmaz, Yazici & Kitsuregawa, 2014) for multimedia information retrieval. The outcomes are reported.

Feature extraction is a simple process in natural language processing that helps users gain a better grasp of the material they are working with for extraction. As soon as the raw text has been cleaned and normalized, it must be transformed into the model’s features. Prior to modeling, users utilize a specific way to give weights to terms in the document. Because computers can easily process numbers, users use word embeddings when representing individual words numerically. The important features are extracted so as to extract the relevant text. The Table 2 and Fig. 6 represent the feature extraction time levels of the proposed and existing models.

Table 2 Feature extraction time levels.

Records considered	Models considered	
	Proposed NDFS-WVM model	Existing BiLSTM-DBN model	Existing RELIEF-MM	
20,000	4	14	12	
40,000	6	20	14	
60,000	8	24	17	
80,000	12	32	21	
100,000	15	35	25	
120,000	16	40	29	

Figure 6 Feature extraction time levels.

Feature extraction is a subset of feature engineering. When data in its raw form is useless, data scientists resort to feature extraction. In order to make use of machine learning methods, feature extraction processes raw data, such as text files, by converting it into numerical features. Data scientists can build new features suited for deep learning applications by extracting the geometry of an object or the similarity in text. The process of selecting features is similar. The process of selecting which features are most likely to improve the quality of prediction variable or output, as opposed to the creation of new features through feature extraction and feature engineering. Feature selection is a method for reducing the complexity of a machine learning model by picking only the most important features to analyze.

Textual data can be transformed into numerical data via feature extraction. NLP feature extraction can be used as an easy way to better comprehend the context in NLP. Textual data must be cleaned and normalized before it can be used in a model since the machine cannot compute textual data. When it comes to computer-processing, numbers are a lot easier to deal with than words. The extraction of features for accurate information extraction and the feature extraction accuracy levels are shown in Table 3 and Fig. 7.

Table 3 Feature extraction accuracy levels.

Records considered	Models considered	
	Proposed NDFS-WVM model	Existing BiLSTM-DBN model	Existing RELIEF-MM	
20,000	95.7	89.6	87.6	
40,000	96	90	88	
60,000	96.4	90.3	88.2	
80,000	96.8	90.6	88.5	
100,000	97	90.8	88.7	
120,000	98	91	89	

Figure 7 Feature extraction accuracy levels.

Methods for scaling independent variables or aspects of data are known as feature scaling. Preprocessing data is also known as normalization and is typically done during this step. The feature normalization helps in balancing the correlated values for better accuracy rate. The Table 4 and Fig. 8 represent the feature normalization accuracy levels.

Table 4 Feature normalization accuracy levels.

Records considered	Models considered	
	Proposed NDFS-WVM model	Existing BiLSTM-DBN model	Existing RELIEF-MM	
20,000	97.2	92.3	91	
40,000	97.5	92.5	91.2	
60,000	97.6	92.8	91.4	
80,000	97.8	93.1	91.5	
100,000	98.1	93.4	91.7	
120,000	98.2	93.5	92	

Figure 8 Feature normalization accuracy levels.

Text categorization includes a step called feature weighting, in which the weight of each feature in a document is calculated. Features are assigned a weight based on how important they are in the classification process, and this method is applied in this research. The more essential a feature is the more weight it receives when it is appropriately weighted. The feature weight allocation accuracy levels of the proposed and existing models are represented in Table 5 and Fig. 9.

Table 5 Feature weight allocation accuracy levels.

Records considered	Models considered	
	Proposed NDFS-WVM model	Existing BiLSTM-DBN model	Existing RELIEF-MM	
20,000	96.8	92.2	90.6	
40,000	97	92.5	90.8	
60,000	97.2	92.7	91.2	
80,000	97.3	93	91.6	
100,000	97.5	93.4	91.8	
120,000	97.8	94	92	

Figure 9 Feature weight allocation accuracy levels.

Feature subsets can be generated using a subset generation approach, a search technology that utilizes sequential and random search algorithms. The users estimate a subset of features based on a set of evaluation methods. The dominant feature subset contains strong weighted features for information extraction. The dominant feature subset generation time levels are shown in Table 6 and Fig. 10.

Table 6 Dominant feature subset generation time levels.

Records considered	Models considered	
	Proposed NDFS-WVM model	Existing BiLSTM-DBN model	Existing RELIEF-MM	
20,000	2	8	17	
40,000	4	12	20	
60,000	5	15	23	
80,000	7	21	26	
100,000	10	25	28	
120,000	12	30	32	

Figure 10 Dominant feature subset generation time levels.

When selecting feature subsets, the goal is to eliminate as much unnecessary and duplicated data as feasible. To make learning algorithms work faster and better, users lower the data’s dimensionality. The Table 7 and Fig. 11 depict the dominant feature subset generation accuracy levels of the existing and proposed models.

Table 7 Dominant feature subset generation accuracy levels.

Records considered	Models considered	
	Proposed NDFS-WVM model	Existing BiLSTM-DBN model	Existing RELIEF-MM	
20,000	97.4	91	92	
40,000	97.6	91.6	92.5	
60,000	97.9	92.3	92.8	
80,000	98.1	93	93	
100,000	98.2	94	93.2	
120,000	98.4	95	93.5	

Figure 11 Dominant feature subset generation accuracy levels.

In computers and information science, information retrieval is the process of acquiring relevant information system resources from a collection of relevant resources. Full-text or other material indexing can be used for searches. Information retrieval is the process of extracting unstructured information, typically in the form of text, from a huge collection of data that is kept on computers. When a user has entered a query into the system, and information is extracted based on the query, this is known as information retrieval. The Table 8 and Fig. 12 represent the information retrieval accuracy levels of the proposed and traditional models.

Table 8 Information retrieval accuracy levels.

Records considered	Models considered	
	Proposed NDFS-WVM model	Existing BiLSTM-DBN model	Existing RELIEF-MM	
20,000	97.8	92	91.5	
40,000	98	92.7	91.7	
60,000	98.1	93	91.8	
80,000	98.3	93.3	92	
100,000	98.5	93.6	92.2	
120,000	98.6	94	92.5	

Figure 12 Information retrieval accuracy levels.

The proposed model extracts all the text features and then applies dominant feature selection that selects the best features from the available feature set. The proposed model allocates weights to the features based on the dependency of feature set. The selected features are used for providing the accurate information extraction based on user query. The proposed model time complexity is reduced as the highly correlated features are selected in the proposed model. The extraction and retrieval time levels of the proposed model are very less than the traditional models because of considering the most useful features with the dominant feature set.

Conclusion

Recent years have seen remarkable developments in deep learning-based image-text cross-modal retrieval systems. Due to the utilization of relations between characteristics and semantics, attributes learned may become a significant trend for retrieval valid technique to its base in adversarial learning & interaction learning. The most pertinent informational images are identified by comparing them to the query image. There have been a variety of philosophical approaches to the problem of efficient and effective feature extraction. One uses measurements of specific attributes as seen by humans and two psychological tests to choose the optimal mathematical model. By keeping these factors in mind, this research analyzed the feature extraction and selection process for the picture retrieval problem. To facilitate feature extraction and selection for information retrieval from huge data utilizing natural language processing models, this research proposed a normalized dominant feature subset with weighted vector model. The inputs and procedures used to select a subset from an original feature set will vary from case to case, resulting in a subset of features that is distinct from the original feature set. In this study, a new set of weighted vector features is introduced and analyzed a feature extraction method. A novel approach is proposed to structural feature extraction using weighted vectors. A fundamental feature selection technique is also proposed for the examination of new features and the development of appropriate features. The proposed model achieves 98.6% accuracy in information retrieval. In future, feature dimensionality reduction models can be used for optimization and for enhanced performance levels. The proposed model can be extended using backward selection technique for the feature dimensionality reduction model. Meta heuristic models can be applied on the proposed model for reducing the feature set.

Supplemental Information

Supplemental Information 1 Source code

Additional Information and Declarations

Competing Interests

Author Contributions

Data Availability

The authors declare there are no competing interests.

Poluru Eswaraiah conceived and designed the experiments, performed the experiments, analyzed the data, performed the computation work, prepared figures and/or tables, authored or reviewed drafts of the article, and approved the final draft.

Hussain Syed conceived and designed the experiments, prepared figures and/or tables, and approved the final draft.

The following information was supplied regarding data availability:

The data is available from Kaggle:

https://www.kaggle.com/datasets/dmaso01dsta/cisi-a-dataset-for-information-retrieval.

The code is available in the Supplementary File.

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
