# Peer review of "Deep learning-based information retrieval with normalized dominant feature subset and weighted vector model"

_PeerJ Computer Science, doi:10.7717/peerj-cs.1805_

## Round 0.1 · original submission · Major Revisions

Please follow the reviewers' comments.

Reviewer 1 ·

Basic reporting

Clear and unambiguous English used throughout.
Organization of the article is not mentioned
The need of proposed method has not been explained at the end of introduction section.
Relevant prior literature is appropriately referenced, but research gap has not been mentioned in literature survey
structure of the article is relevant to the journal format
No satisfactory in the comparison of results. Authors are not compared with multiple recent techniques.

Experimental design

The submission clearly defines the research question.
Some of the diagrams not specific to the proposed approach.

Validity of the findings

There is a no novel approach in this research.
Accuracy level has been compared with only one method that also very light variation with proposed approach.
The conclusions are not appropriately stated.

Additional comments

There is no novelty in their findings.•
All the references are not cited
No uniformity in their references.
No satisfactory level in the result discussion
As far as your future prospects are concerned, You may specify the way You improve Your model and techniques

·

Basic reporting

The manuscript presents a comprehensive review of relevant literature in information retrieval, mainly focusing on semantic-based heterogeneous multimedia retrieval and various methods to enhance retrieval performance. The paper provides a succinct overview of each discussed method, highlighting their strengths and contributions to the field. The literature survey is well-structured, allowing readers to follow the progression of discussed methods and their contributions to the overall topic of the manuscript.
While the manuscript demonstrates a comprehensive understanding of the proposed model, there are areas where could be improved. For example, I suggest that the figures should generally have the quality shown in Figures 1 and 2 in terms of resolution, especially Figure 5.

Experimental design

The manuscript presents a comprehensive exploration of the proposed model for information retrieval. The manuscript delves into the importance of feature extraction and the advantages of deep learning techniques. It compares the proposed Normalized Dominant Feature Subset with the Weighted Vector Model (NDFS-WVM) with the traditional BiLSTM-DBN model. The manuscript provides an in-depth description of the various steps involved in the proposed NDFS-WVM, including feature extraction, normalization, weight allocation, and dominant feature subset generation. This detailed exposition aids in comprehending the complexity of the model's workflow.
It would be interesting to cite a reference to the bidirectional model Long Short Term Memory based Deep Belief Network (BiLSTM-DBN) or add a brief explanatory note in Lines 315-317.

Validity of the findings

After presenting the results of the empirical comparison, the authors could interpret the implications of these findings. For example, How does the performance of NDFS-WVM compared to BiLSTM-DBN impact the field of information retrieval?

·

Basic reporting

The description is not clear for understanding by the reader.

Figurers need to be explained in detail.

Block diagram is needed for the proposed approach with detailed explanation.

Experimental design

The experiment needs proper description.

Description about the methodology need more details.

Validity of the findings

Results need to be compared with the existing approaches.

---

## Round 0.2 · Major Revisions

Please follow the third reviewer's comments.

Reviewer 1 ·

Basic reporting

Clear and unambiguous English used throughout.
Relevant prior literature is appropriately referenced.
The structure of the article conforms to the acceptable format. Figures are relevant to the content of the article.

Experimental design

The submission clearly defines the research question.
Methods are described with sufficient information.

Validity of the findings

There is a novel approach in this research.
The data is robust and statistically sound.
The conclusions are appropriately stated.

Additional comments

All the references are not in uniform style.

·

Basic reporting

The manuscript presents a comprehensive review of relevant literature in information retrieval, mainly focusing on semantic-based heterogeneous multimedia retrieval and various methods to enhance retrieval performance. The paper provides a succinct overview of each discussed method, highlighting their strengths and contributions to the field. The literature survey is well-structured, allowing readers to follow the progression of discussed methods and their contributions to the overall topic of the manuscript.

Experimental design

The manuscript presents a comprehensive exploration of the proposed model for information retrieval. The manuscript delves into the importance of feature extraction and the advantages of deep learning techniques. It compares the proposed Normalized Dominant Feature Subset with the Weighted Vector Model (NDFS-WVM) with the traditional BiLSTM-DBN model. The manuscript provides an in-depth description of the various steps involved in the proposed NDFS-WVM, including feature extraction, normalization, weight allocation, and dominant feature subset generation. This detailed exposition aids in comprehending the complexity of the model's workflow.

Validity of the findings

The article presents results relevant to the field of information retrieval, as well as briefly discussing the comparison with other methods. The data shown is an excellent way to make the research reproducible.

·

Basic reporting

Authors used deep learning based information retrieval system for extracting the text information. Normalized Dominant Feature Subset with Weighted Vector Model (NDFS-WVM) is proposed for feature extraction and selection for information retrieval and reported an accuracy of 98.6%. But the manuscript needs to be improved in the following points for better understanding.

1. Abstract and conclusion need to be modified precisely based on the novelty and the outcomes.

2. In literature review only five papers are mentioned, which are published in 2023, 2022, 2021, and 2020. It will be good if some more papers will be compared related to the state of art. The contains discussed in literature review need to be organized in tabular format (mentioning, year, technique, database, limitations, and the remarks) for better understanding by the reader.

3. Caption of the figures need to be given.

4. The block diagram needs detailed explanation.

5. Figure 3 represents the hidden layer representation. What exactly this is conveying? The clear information about the hidden layers is missing (based on the number of neurons as well as the layer information).

6. Figure 4 represents the weighted vector representation. How these weights are assigned? It needs to be discussed more in detail.

7. Proposed method individual step need to be discussed more in detail with justification.

8. Figure 6 represents the number of features computation for different models w.r.t. no of records . It shows the less number of features for NDFS-WVM model for same number of text record. It will be better if this may be represented in form of table as the number of records are discrete and not continuous. Same can be followed for figure 7 to figure 12.

9. Table 9 is not clear. What these values representing? Not mentioned in the table or text. Same is for Table 2 to Table 6.

Experimental design

The detail information about the model to be represented with justification. Feature extraction need to be explained more detail.

Validity of the findings

Results: Need modification and to be discussed more in detail. Database is missing

Additional comments

Through checking is required.

Proper organization is required.

---

## Round 0.3 · accepted · Accept

The authors revised and improved the paper significantly.

·

Basic reporting

Authors have addressed the queries raised by the reviewers.
The manuscript may be considered for publication

Experimental design

Well explained.

Validity of the findings

Well elaborated.